# Electrical Characterization of Cellulose-Based Membranes towards Pathogen Detection in Water [note 1]

**DOI:** 10.3390/bios11020057

**Published:** 2021-02-21

**Authors:** Grégoire Le Brun, Margo Hauwaert, Audrey Leprince, Karine Glinel, Jacques Mahillon, Jean-Pierre Raskin

**Affiliations:** 1Institute of Information and Communication Technologies, Electronics and Applied Mathematics, UCLouvain, 1348 Louvain-la-Neuve, Belgium; margo.hauwaert@student.uclouvain.be (M.H.); jean-pierre.raskin@uclouvain.be (J.-P.R.); 2Laboratory of Food and Environmental Microbiology, Earth and Life Institute, UCLouvain, 1348 Louvain-la-Neuve, Belgium; audrey.leprince@uclouvain.be (A.L.); jacques.mahillon@uclouvain.be (J.M.); 3Institute of Condensed Matter and Nanosciences (Bio and Soft Matter), UCLouvain, 1348 Louvain-La-Neuve, Belgium; karine.glinel@uclouvain.be

**Keywords:** paper-based sensors, nitrocellulose, impedance measurements, dielectric properties, parallel-plate electrodes, interdigital electrodes, endolysins, *Bacillus thuringiensis*

## Abstract

Paper substrates are promising for development of cost-effective and efficient point-of-care biosensors, essential for public healthcare and environmental diagnostics in emergency situations. Most paper-based biosensors rely on the natural capillarity of paper to perform qualitative or semi-quantitative colorimetric detections. To achieve quantification and better sensitivity, technologies combining paper-based substrates and electrical detection are being developed. In this work, we demonstrate the potential of electrical measurements by means of a simple, parallel-plate electrode setup towards the detection of whole-cell bacteria captured in nitrocellulose (NC) membranes. Unlike current electrical sensors, which are mostly integrated, this plug and play system has reusable electrodes and enables simple and fast bacterial detection through impedance measurements. The characterized NC membrane was subjected to (i) a biofunctionalization, (ii) different saline solutions modelling real water samples, and (iii) bacterial suspensions of different concentrations. Bacterial detection was achieved in low conductivity buffers through both resistive and capacitive changes in the sensed medium. To capture *Bacillus thuringiensis*, the model microorganism used in this work, the endolysin cell-wall binding domain (CBD) of Deep-Blue, a bacteriophage targeting this bacterium, was integrated into the membranes as a recognition bio-interface. This experimental proof-of-concept illustrates the electrical detection of 10^7^ colony-forming units (CFU) mL^−1^ bacteria in low-salinity buffers within 5 min, using a very simple setup. This offers perspectives for affordable pathogen sensors that can easily be reconfigured for different bacteria. Water quality testing is a particularly interesting application since it requires frequent testing, especially in emergency situations.

## 1. Introduction

Access to safe and sufficient water is a prerequisite for to the development of human communities and the enhancement of economic activities [1]. Identification, control and prevention of ground and surface water pollution require both frequent water quality testing and diligent water management. The detection of pathogenic bacteria such as *Escherichia coli*—used as an indicator for fecal water contamination—usually deploys techniques that include colony count, DNA analysis after polymerase chain reaction (PCR) or enzyme-linked immunosorbent assay (ELISA) [2,3]. Despite the favorable performance (detection limits of 1 CFU mL^−1^) of these techniques, their high cost, their requirement of well-equipped laboratory facilities, and the time constraint (at least several hours per test) have prompted the development of portable, simple and low-cost tools suitable for a rapid (less than 1 h) and precise detection methods of pathogens. Furthermore, the sanitary pandemic caused by the infectious severe acute respiratory syndrome coronavirus 2 (SARS-CoV-2) highlighted the field of action of point-of-care (PoC) devices because it can meet the need of mass screening, in particular to screen the virus in water environments. Wastewater-based epidemiology (WBE) is a promising approach to predict the potential spread of the infection by testing for infectious agents in wastewater, and has been approved as an effective way to obtain information on diseases and pathogens [4,5].

Microfluidic paper analytical devices (μPADs), on the one hand, are the dominant PoC biosensors for the rapid detection of pathogens in both healthcare and environmental monitoring, especially in situations with scarce resources [6,7]. Given the accessibility of this technology, it is appropriate for either water scientists or citizen groups, without requiring specific training [8]. Paper is a valuable platform for biodetection as it presents several assets. First, it has beneficial spontaneous microfluidic properties through capillarity. Second, it facilitates the attachment of bioreceptors which are often proteins such as antibodies, that contribute to specificity towards pathogens. Third, it is low-priced and it allows for straightforward manufacturing and disposability. These benefits are already used in lateral flow assays (LFA) such as pregnancy test strips, in which analytes of interest are passively drained through a nitrocellulose (NC) membrane, a paper derivative, towards the detection zone where they are immobilized by specific bioreceptors. Current paper-based sensors mostly require the use of labels (such as gold nanoparticles, which have an intense red color), conjugated with antibodies to achieve specific optical detection of the immobilized analytes [9]. However, they have two main disadvantages that inhibit their use in the field of water potability, which has demanding limits of detection for pathogens and pollutants. Optical LFA have a reduced sensitivity since only the top 10 μm depth of the paper contributes to the colorimetric signal due to the opaqueness of the membrane [10]. Additionally, the measurement result is mostly qualitative or semi-quantitative.

Electrical biosensors, on the other hand, rely on the monitoring of changes in material electrical properties when bacteria bind in close proximity from the surface of, e.g., interdigital electrodes (IDE) designed on a solid substrate [11,12]. The signal response, often proportional to the number of bacteria, is used as an electrical fingerprint of the sample to provide fast, precise and quantified information about the bacteria presence in water. However, the particular mechanisms of electron transfer between electrodes and specific bacterial cells, as well as within the cells, are still under fundamental studies in bioelectrochemistry [13]. Furthermore, the production and usage of electrical biosensors face significant obstacles. Grafting a biorecognition layer, e.g., bacteriophages (phages) or antibodies, on the surface of conventional surface-based electrical biosensors faces problems such as reproducibility, uniformity and stability over time [14]. The functionalization protocol needs to be adapted to every surface material and grafting molecule. In addition, the capture percentage of bacteria by the biorecognition layer is relatively low since only bacteria in close vicinity to the surface bind to the specific receptors. Many of the target pathogens thus flow over the electrode without binding, decreasing the sensor sensitivity. Finally, conventional surface-based electrical biosensors utilize gold electrodes functionalized with bioreceptors/antibodies using classical thiol chemistry [15,16]. However, insufficient chemical stability of thiolates is one of the most serious problems for their applications in ambient and aqueous environments [17].

Despite the aforementioned drawbacks of both individual sensing technologies, studies have shown that the integration of highly sensitive electronic detection methods with LFA is an attractive approach to circumvent these and capitalize on the advantages of paper substrates and electrical biosensors [18]. Given the favorable electrostatic properties of nitrocellulose, bioreceptors are readily immobilized through the whole pore volume of the paper membrane, thus drastically increasing the number of interactions with targeted pathogens. Unlike surface-based methods, electrical measurements taking advantage of paper porosity thus allow to quantify the number of bioreceptor-bacteria conjugates in the whole tested sample volume. However, the development of such sensors remains very challenging for three main reasons.

First, one of the key factors affecting the analytical performance of μPADs is the bioreceptor used to capture the bacteria in the test zone. Antibodies, commonly used as a biointerface in μPADs, are rather expensive. As a result, there is a growing interest in developing proteins as alternative receptors for LFA. Particularly promising are bacteriophages, viruses that specifically infect bacteria and produce lytic enzymes called endolysins that show strong affinity and high specificity towards target bacteria [19,20].

Second, due to significantly different properties of paper-based and more conventional substrates for electronic circuits, innovative design methods are needed. Alternative manufacturing techniques, e.g., printing sensors such as IDE, are being explored as integrated sensing devices [21,22,23]. However, integrating electronics on paper substrates is difficult to implement because of the inhomogeneous nature of the paper, resulting in low electrode resolutions (~hundreds of μm) and high electrical resistances [24] with respect to the classical design of microelectrodes suitable for bacteria sensing (a few μm finger gaps) [25].

Third, prior works that attempted to accommodate electrical bacteria detection on common μPADs were mostly based on direct charge transfer measurements [26,27]. However, these require cumbersome redox probes. Furthermore, electrical measurements are influenced by the ionic strength of the analyzed solution. Direct current measurements are particularly affected by this ionic background noise, since they only measure the solution resistance, inversely proportional to the number of ions. When dealing with aqueous samples presenting varying electrical conductivities, therefore, it is challenging to calibrate the sensor and to differentiate the electrical response of target compounds from the electrical background signal.

In this paper we address the aforementioned challenges by emphasizing three contributions towards paper-based electrical biosensing for simple, rapid and affordable bacteria detection in water.

First, we capitalize on the natural capillarity of the NC membrane to wick bacterial suspensions to the testing zone, where the membrane is functionalized with the recently characterized cell-wall binding domain (CBD) derived from the PlyB221 endolysin encoded by phage Deep-Blue targeting *Bacillus thuringiensis* [28], used as model microorganism in this work. This ensures high binding and immobilizing capacity towards this specific bacteria.

Second, we demonstrate the relevance of the simple parallel-plate setup presented in [29]. This common material dielectric measurement system can be judiciously used as sensor to perform electrical measurements on NC membranes inserted in between the electrodes. This plug-and-play setup eliminates the unpractical need to integrate electronics components directly onto the paper matrix.

Third, impedance sensing is proposed to monitor the porous NC membrane permittivity changes caused by various electrolyte solutions. Since no electron transfer occurs at the electrode surface, changes in electrical properties are mainly observed through volumic impedance properties. By measuring the later with an AC signal, both resistive and capacitive properties are estimated at high frequencies, which enables the label-free and non-intrusive detection of bacteria immobilized by specific bioreceptors. Indeed, their presence in the membrane affects both the global conductivity and permittivity [25,30].

The remainder of this paper is organized as follows. We began by validating the CBD biointerface, both electrically and optically. In order to characterize the system behavior and ionic noise for aqueous solutions with different ionic strengths, we then analyzed the sensor response and resolution to saline solutions at different salt concentrations, used as models of real aqueous samples. An equivalent electrical model of the sensing system was developed to quantify the impact of ionic concentration on the total measured impedance. Then, the sensing principle was validated in the presence of bacterial cells. A proof of concept of the simple and rapid (<5 min) parallel-plate biosensor was demonstrated by detecting *B. thuringiensis* cells in low-conductive buffers. Finally, the bacterial detection results with the plug-and-play parallel-plate setup were compared with a planar fringing field electrodes system, composed of IDE directly applied on a single side of the NC membrane. The sensing principles of both parallel-plate and IDE devices were modeled and analyzed using small-signal electrical equivalent circuits, highlighting the contribution of ions in both bacterial detection mechanism. Their potentials, advantages and limitations are also discussed.

## 2. Materials and Methods

### 2.1. Materials

Nitrocellulose membranes on a polyester backing (UniSart Lateral Flow CN95 Backed, 20 µm nominal pore size) were purchased from Sartorius (Göttingen, Germany). Phosphate-buffered saline (PBS) and sodium chloride solution (NaCl, 1 M) were purchased from Sigma-Aldrich (St. Louis, MO, USA). Deionized (DI) water was produced in our facilities (conductivity σ = 6.6 × 10^−6^ S/m).

### 2.2. Biological Procedures

#### 2.2.1. Bacterial Strains and Growth Conditions

*B. thuringiensis GBJ002 expressing the cyan fluorescent protein (CFP)* was used as proof-of-concept for this study. Bacteria were plated on lysogeny broth (LB)-agar plate containing kanamycin (50 µg/mL) and acid nalidixic (25 µg/mL) and grown overnight (O/N) at 30 °C. Next, an individual colony was used to inoculate 5 mL of LB supplemented with antibiotics and the suspension was incubated O/N at 30 °C with agitation (120 rpm). Then, the culture was washed twice with an appropriate buffer (water or diluted PBS) to remove residual LB (centrifugation at 10,000× *g* for 5 min at room temperature (RT)). The pellet was finally re-suspended in the appropriate buffer yielding a bacterial suspension of ca. 10^8^ CFU mL^−1^.

#### 2.2.2. Phage Endolysins Expression and Purification

A detailed description of the expression and purification of the CBD of PlyB221 endolysin, encoded by phage Deep-Blue can be found in [28]. The CBD was fused to a green fluorescent protein (GFP) for fluorescence assays. The protein concentration was adjusted to 1 mg/mL.

#### 2.2.3. Preparation and Characterization of the Cell-Wall Binding Domain (CBD) Biointerface for Specific Bacteria Capture

A protocol for optimal biofunctionalization of the NC membranes with endolysin CBD based on simple physisorption was developed (Figure 1). Membranes were prepared by individually depositing and shaking 50 µL of 1 mg/mL solution of purified CBD on the NC. Membranes were then dried in an oven for 60 min at 37 °C followed by desiccation for 30 min at RT to fix the proteins to the membrane. Following desiccation, membranes were washed twice during 2 min with deionized (DI) water to remove proteins in excess. Membranes were wiped, dried, and finally stored in a desiccation chamber at RT. Effective binding of specific bacteria to the biointerface was assessed by depositing a droplet of *B. thuringiensis* suspension on the biofunctionalized membranes followed by a washing step (Figure 1). Confocal laser scanning microscopy (Zeiss LSM 710) experiments were performed to observe bacteria immobilization in the membrane depth.

### 2.3. Parallel-Plate Setup

#### 2.3.1. Impedance Sensing

Figure 2A shows a schematic of the experimental system for the simple parallel-plate measurements. The nitrocellulose membranes were held between the two parallel-plates of the dielectric test fixture (16451B, Agilent, Santa Clara, CA, USA), connected to an impedance analyzer (LCR 4284A, Agilent, Santa Clara, CA, USA). The impedance spectroscopy measurements were carried out with the LCR, remotely controlled by a computer through the Labview software (Labview National Instrument, Austin, TX, USA) to perform an automatic sweep from 1 kHz to 1 MHz, at voltage amplitude of 200 mV. Before impedance measurement, an open-circuit calibration was performed. The impedance data were extracted in a magnitude-phase data-structure. The ZVIEW software (Scribner Associates Inc., Southern Pines, NC, USA) is used to fit the electrical equivalent model of the parallel-plate sensing system, as presented in Figure 2A, and extract the model parameter values from the measurements.

The material real and imaginary dielectric constant are derived for a given membrane thickness by measuring its capacitance and dissipation factor. An electrical model is used to differentiate the actual dielectric properties of the NC from those of the polyester backing (Figure 2A).

#### 2.3.2. Sensor Modelling

An analytical model of the NC membranes soaked with biological solutions and inserted in between the two parallel-plate electrodes is established to provide a physical understanding of the system. In the presence of an electrolyte, the equivalent electrical model considers a double layer capacitance (*C_dl_*). The equivalent circuit also considers the NC membrane, modelled through the parallel association of its capacitive (*C_NC_*) and conductive (*R_NC_*) properties, as well as the polyester backing, modelled using its capacitive properties only (*C_Backing_*), since its dielectric losses are considered to be negligible in the studied frequency range. The value of the polymer backing capacitance (~100 pF) is determined experimentally through measurement of its permittivity by means of dielectric measurements (see Section 2.3.1). *C_NC_* and *R_NC_*, representing the electrical properties of the NC membranes saturated with saline solutions or bacterial suspensions, strongly depend on the ionic concentration in the solution. The double layer capacitance, representing the interfacial properties is given by [19]:(1)$$C_{dl}=\frac{\varepsilon_{0}\varepsilon_{r,sol}}{\sqrt{\frac{\varepsilon_{0}\varepsilon_{r,sol}k_{B}T}{2\;q^{2}N_{av}c_{ions}10^{3}}}}A_{e}$$
with *C_dl_* the double layer capacitance, *k_B_* the Boltzmann constant, *c_ions_* molar ionic concentration of the solution in which the double layer occurs, *ε_r,sol_* the relative permittivity of this solution and *A_e_* the surface of the parallel-plate electrode. The values for *C_dl_* at different ionic concentrations were simulated using COMSOL Multiphysics v.5.4 (COMSOL AB, Stockholm, Sweden) based on Equation (1), and lie in the 10–100 µF range. Regarding the values of the other parameters, their influence can therefore be neglected at the considered frequencies between 1 kHz and 1 MHz.

The equivalent model has three cut-off frequencies:(2)$$f_{1}=\;\frac{C_{dl}+C_{Backing}}{2\pi \cdot R_{NC}\cdot \left[C_{dl}C_{Backing}+C_{NC}\left(C_{dl}+C_{Backing}\right)\right]}$$
(3)$$f_{2}=\;\frac{C_{dl}\;+\;C_{Backing}}{2\pi \cdot C_{dl}C_{Backing}}$$
(4)$$f_{3}=\;\frac{1}{2\pi \cdot R_{NC}C_{NC}}$$

### 2.4. Interdigital Electrodes (IDE) Setup

#### 2.4.1. Interdigital Electrode Design and Fabrication on Nitrocellulose (NC) Membranes

The deposition of IDE on nitrocellulose substrate was conducted using a physical vapor deposition (PVD) e-gun evaporation technique. Gold IDE were deposited by applying patterned nickel masks on top of the nitrocellulose membrane. This deposition technique is more cumbersome than screen-printing and inkjet printing techniques, which are usually used for deposition of electrodes on paper [31]. However, these have the disadvantage of not allowing precise electrode deposition on chemically untreated nitrocellulose. Hence, PVD is chosen because it allows for precise deposition without inducing variability through additional treatment. The IDE finger width and interdigit gap are 200 µm, which enables the detection of dielectric properties over the whole NC membrane depth (~140 µm) [32].

#### 2.4.2. IDE Impedance Sensing

Figure 2B shows a schematic of the experimental system for IDE measurements. The IDE were connected through toothless crocodile clips to an impedance analyzer (LCR 4284A, Agilent, Santa Clara, CA, USA) through BNC connectors. The impedance spectroscopy measurements were carried out with the LCR, remotely controlled by a computer through the Labview software (Labview National Instrument, Austin, TX, USA) to perform an automatic sweep from 1 kHz to 1 MHz, at voltage amplitude of 20 mV. Before impedance measurement, an open-circuit calibration was performed without any electrical contacts between the crocodile clips. The impedance data were extracted in a magnitude-phase data-structure.

#### 2.4.3. IDE Sensor Modelling

The equivalent circuit of the IDE in Figure 2B incorporates the surficial phenomenon of double layer capacitance through *C_dl_* and the volumic phenomena through *C_air_*, *R_air_*, *C_NC_*, and *R_NC_*, corresponding to the upper air layer and lower nitrocellulose layer, respectively. Given the width of IDE fingers, the backing is not taken into account (Section 2.4.1). The double layer capacitance for IDE electrodes in contact with a given solution is extended from (1) to:(5)$$C_{dl}=\frac{\varepsilon_{0}\varepsilon_{r,sol}}{\sqrt{\frac{\varepsilon_{0}\varepsilon_{r,sol}k_{B}T}{2\;q^{2}N_{av}c_{ions}10^{3}}}}A_{e}\left(N-1\right)$$
with *A_e_* the surface per electrode finger and *N* the number of fingers. The equivalent resistance and capacitance of the nitrocellulose and air volume are given by
(6)$$\begin{array}{cc} R_{NC}=K_{cell}\sigma_{NC}^{-1} & R_{air}=K_{cell}\sigma_{air}^{-1}\\ C_{NC}=K_{cell}^{-1}\varepsilon_{0}\varepsilon_{r,NC} & C_{air}=K_{cell}^{-1}\varepsilon_{0}\varepsilon_{r,air} \end{array}$$
with *K_cell_* the cell constant, *ε_r_* and σ the relative permittivity and conductivity of the sensed nitrocellulose and air volumes, respectively. The cell constant is determined experimentally and incorporates the geometric properties of the IDE [33]. Hence, it does not vary with frequency nor with the electrical properties of the material.

### 2.5. Sensing of Saline Solutions as Models for Real Water Samples

Prior to electrical measurements, sodium chloride solutions of various concentrations (*c_NaCl_*), 10^−5^, 10^−4^, 5 × 10^−4^, 10^−3^, 5 × 10^−3^, 10^−2^, 10^−1^ mol/L (M), were prepared by dilutions of a 1 M NaCl solution in DI water, in order to model the electrical properties of different types of real water sample and biological buffer (Table 1). Regarding the impedance measurements considered in this study, the parameters of interest to be modelled are the mean ionic strength of the liquid (through adjustment of c*_NaCl_*), and hence its dielectric properties (electrical conductivity and permittivity).

Then, 50 µL of the diluted sodium chloride solutions were then deposited on NC membranes, placed in between the parallel-plate electrodes, and reference dielectric or impedance measurements were performed within 5 min. The parallel-plates were wiped between each measurement to remove remaining water and salt on the electrodes.

### 2.6. Bacteria Detection in Physiological Buffers

Before depositing bacterial suspensions, 50 µL of PBS buffer diluted 1000× (PBS:1000) in DI water was deposited on a previously biofunctionalized membrane, and reference dielectric or impedance measurements were performed within 5 min with parallel-plates or IDE, respectively. PBS:1000 was chosen as biological buffer because the largest detection sensitivities were shown to be achieved with low-salt buffer solutions [25], and such low-salt buffers have electrical properties similar to real water samples. Five min is a time limit before which it is assumed that the wet impedance has not changed by more than 5% due to drying. Then, suspensions of 10^8^, 10^7^ and 10^6^ CFU mL^−1^ of stationary-state *B. thuringiensis* resuspended in PBS 1:1000 were deposited (50 µL) on top of the membrane sample with a micropipette (Figure 2A,B) and spread within the membrane due to capillarity, and impedance or dielectric measurements were performed. In order to observe the global sensitivity of the setup, including both sensitivities of the impedance modulus and phase to bacteria presence, the sensitivities (*S*) in the explored frequency range were computed as the amplitude of the differences in complex impedance measured with and without bacterial cells, in percent:(7)$$S=\left|\frac{Z_{Bact}-\;Z_{PBS}}{Z_{PBS}}\right|$$

## 3. Results

### 3.1. Characterization of the CBD-Biofonctionalized Nitrocellulose Membrane

#### 3.1.1. Optical Characterization of the CBD Biointerface

In this section, we validate the biofunctionalization of the NC membrane with a CBD-biointerface aimed at capturing *B. thuringiensis* whole cells for subsequent electrical detection. The binding of CBD to bacterial cells was evaluated in a cell wall decoration assay as described by [28], relying on the homogeneous adsorption of GFP-CBD to *B. thuringiensis* cells observed by fluorescence microscopy. This confirms their potential as specific immobilization probes for LFA biosensor schemes. In [36], we observed that deposited specific proteins (antibodies) were completely and uniformly distributed throughout the thickness of the NC membrane, promising the capture of bacterial cells throughout the volume and thus enabling electrical detection over the whole volume. In this paper we demonstrate that, by applying the developed protocol, GFP-CBD has been successfully deposited over the whole NC volume (Figure 3). Confocal microscopy images captured after deposition of the bacterial suspension and subsequent washings of the membrane also demonstrates strong affinity and robust capture of *B. thuringiensis* by the CBD within the NC, confirming the potential of the CBD as an immobilized probe in paper detection schemes. Furthermore, the porous structure of the nitrocellulose membrane is highlighted by the biofunctionalization of the substrate with fluorescent bioreceptors which experimentally confirms the mean pore size of the membrane (around 20 µm). The size of *B. thuringiensis* cells, about 0.5–1.0 µm × 2–5 µm [37], and their tendency to form aggregates [38], justifies this pore diameter since clogging and retention to the membrane should be avoided.

#### 3.1.2. Electrical Characterization of Dry and Biofunctionalized Nitrocellulose Membranes

In order to characterize the impact of the biofunctionalization on the membrane electrical properties, we performed dielectric measurements with the parallel-plate electrodes on raw and CBD-biofunctionalized NC membranes, under dry conditions. The permittivity of a raw nitrocellulose membrane drops from 1.55 to 1.45 between 1 kHz and 1 MHz (Figure 4). The biofunctionalization causes a permittivity reduction of approximatively 1–3%.

The shaded area around the measurement curves in Figure 4 express the standard errors bars over the frequency range of interest, and indicate that results show variability to a certain extent, which can be explained by two parameters. First, the NC membranes present a foam-like structure (Figure 3) resulting in structural anisotropy and surface inhomogeneity, which renders absolute measures of the permittivity difficult as the solver algorithm assumes that the material under test is homogenous [39]. Second, the biofunctionalization protocol can substantially modify the permittivity of the NC sheets by altering its pore surface properties. Indeed, under conditions of low humidity reached after the desiccator step in the protocol, nitrocellulose membranes accumulate a significant static charge [9], affecting the dielectric measurements.

### 3.2. Impact of the Electrolyte Conductivity on the Parallel-Plate Sensor Response

Table 2 summarizes the observed changes in the equivalent circuit elements of Figure 2A for different NaCl concentrations. Between 1 kHz and 1 MHz, we observe that the impedance measurements are particularly sensitive to the electrical conductivity (*σ_sol_*) of the electrolyte, as it directly influences *R_NC_*. Thus, to understand the response of the parallel-plate sensing system and discriminate the bacteria electrical contribution, it is of utmost importance to characterize the effect of the background ionic noise resulting from remaining dissolved salt in the solution. Not monitoring or controlling the ionic concentration of the electrolyte where bacteria are suspended could lead to misinterpretation of the electrical results, as the supposed detection of elements could be caused by changes in the background ionic strength.

The concentrations of the saline solutions were chosen to represent background ionic noise for different water sources of interest (Table 1). Impedance measurements were first carried out to investigate the impedance modulus and phase dependence upon the salt concentration between 1 kHz and 1 MHz. Figure 5A,B show that the sensor discriminates the different saline concentrations through shifts of both impedance magnitude and phase. A global decrease of the impedance magnitude is observed with increasing NaCl concentration, since increased salt concentration increases the conductivity *σ_sol_*. This magnitude decrease is higher between 10 kHz and 1 MHz, where the impedance phase is mostly resistive: this is where the highly varying *R_NC_* has the most effect on the total complex impedance. Even if these peaks tend towards resistive impedance angles, they stay lower than −45° given that no direct conduction path exists between the two parallel electrodes due to the isolating backing (Figure 2A). At both sides of these peaks, the phase decrease indicates a transition from a mixed to an exclusively capacitive behavior, led by the volume capacity C_NC_ at higher frequencies. We observe a frequency shift of the peak, shifting towards higher frequencies when the salt concentration increases. This can be explained by Equations (2) and (4): *f_1_* and *f_3_* correspond to the middle of the upwards and downwards flank of the peak. These cut-off frequencies increase when the salt concentration increases, given the variation of *R_NC_* and *C_NC_* in Table 2.

The working frequencies of the sensor towards detection of saline solutions corresponding to ionic noise of interest (Table 1) lies in the range 10–200 kHz. The limit of detection of the system (LOD) lies between 10^−5^ M and 10^−4^ M since the sensing device was not able to differentiate significantly (<3σ) impedance modulus and phases. This LOD corresponds to very low salinity electrolytes, less conductive than most of the buffers considered in biological detection schemes, which is beyond the scope of interest for the sensor-applications.

In order to quantify the impact of changes in salt concentration on the global system impedance, we extracted the values of *R_NC_* and *C_NC_* for the different salt concentrations based on the simple electrical equivalent model of the parallel-plate setup (see Appendix A for the data). The system is sensitive to both resistive and capacitive effects, even though, comparatively, the increasing ionic strength of the solution is more sensed through *R_NC_* than *C_NC_* (Table 2). In addition, the quantitative evaluation of *R_NC_* and *C_NC_* also supports the LOD of 10^−4^ M, as the difference in impedance, resistance and capacitance with the 10^−5^ M solution is less than 3 times the standard deviation (σ).

Dielectric measurements were carried out to corroborate the impedance measurements. Relative permittivity of NC membranes soaked with 10^−4^ M and 10^−3^ M NaCl solutions, modeling respectively highly diluted PBS (PBS:1000) and slightly saline solutions, was extracted over the frequency range of interest (Figure 5C). The permittivity of the saturated membrane with backing shows an increase of the system permittivity with the salt concentration. This is reflected by an increasing value of *C_NC_* extracted from the impedance measurements (Table 2), even if the proportions are different since this system permittivity considers also the double layer and backing permittivity while *C_NC_* only incorporates the volume of the nitrocellulose.

### 3.3. Detection of B. thuringiensis Cells with the Parallel-Plate Setup

After the characterization of the sensor response to ionic background noise in aqueous solutions, impedance measurements of *B. thuringiensis* resuspended in the low-salt buffer PBS:1000 was investigated to extend the electrical model assessed in Section 2.3 to the detection of label-free, whole bacterial cells. As the impedance measurements are extremely sensitive to the electrical conductivity *σ_sol_* of the electrolyte, it is almost impossible without labels to directly predict the bacterial concentration from a single measure since two samples with identical bacterial loads but different conductivities would result in different signals. Effective discrimination of the electrical footprint of the bacterial cells from the ionic background noise thus requires comparing the sensor signal to an appropriate control value.

Therefore, we performed differential measurements, comparing the signal obtained with samples of 10^8^, 10^7^ and 10^6^ CFU mL^−1^ of *B. thuringiensis* to blank PBS:1000 measurements (Figure 6A). Significant differences with and without bacterial cells were observed between 10 kHz and 1 MHz for both impedance modulus and phase, where the membranes soaked with bacterial solutions show a lower value of impedance modulus than the blank reference PBS:1000 buffer. The measurements for bacterial cells and PBS:1000 follow the same typical decrease of impedance modulus than observed for low salinity solutions (Figure 5A). The phase peak of the membrane filled with 10^8^ CFU mL^−1^ bacterial suspension (Figure 6B) is also subjected to a shift towards higher frequencies. The bacteria and PBS:1000 impedance modulus and phase curves show good fitting with the impedance modulus and phase of the NaCl solutions. In particular, PBS:1000 impedance curves lie in between the impedance curves of 2 × 10^−4^ M and 5 × 10^−4^ M NaCl solutions, while the impedance of the solution containing the 10^8^ CFU mL^−1^ bacterial suspension lies between the 5 × 10^−4^ M and 10^−3^ M curves. Regarding the PBS:1000 buffer, the approximate correspondence of its dielectric properties to the range of 2–5 × 10^−4^ M NaCl solution is confirmed as the salt concentration used to model this buffer is precisely 1.6 × 10^−4^ M (Table 1). To deepen the comparison between the bacterial and saline solutions, we have extracted the membrane resistance *R_NC_* and capacitance *C_NC_* of the electrical model under the PBS:1000 and 10^8^ CFU mL^−1^ bacteria conditions (see Appendix A for the data). *R_NC_* and *C_NC_* extracted for the PBS:1000 condition diverges of around 5% from the 5 × 10^−4^ M NaCl condition, against about 10% between the solution containing 10^8^ CFU mL^−1^ bacteria and the 10^−3^ M NaCl model solution. For their part, the 10^7^ and 10^6^ CFU mL^−1^ bacterial suspensions show a significant drop in impedance modulus (Figure 6A, insert) relative to the PBS:1000 buffer. However, unlike the 10^8^ CFU mL^−1^ suspensions, the shift in the phase peak is not significant for the 10^7^ and 10^6^ CFU mL^−1^ bacteria solutions (Figure 6B) and. therefore, cannot be used to assess the bacteria presence in the membrane. In addition, the impedance modulus and phases of 10^7^ and 10^6^ CFU mL^−1^ suspensions are overlapping over the whole spectrum. This suggests an intrinsic limit of detection of the parallel plate towards bacteria detection of about 10^7^ CFU mL^−1^.

Dielectric measurements were carried out to substantiate the bacterial detection results through impedance measurements (Figure 6C). Here again, the relative shift between PBS:1000 with and without bacteria follows a similar tendency to the model saline solutions. This tends to confirm the similarities in dielectric property variations between solutions with and without bacteria, and the differences in salt concentration. It is thus consistent to pose the hypothesis that bacteria are sensed through increase in ion concentration.

### 3.4. Comparison with Another System: B. thuringiensis Detection with the IDE Setup

In order to assess the detection results obtained with the plug-and-play parallel-plate system where the electrodes are deported, we considered to deposit metallic IDE directly on top of the nitrocellulose membrane to reach an integrated sensing device.

#### 3.4.1. Gold IDE Deposited on Nitrocellulose Membranes

Au-IDE were successfully deposited on top of the NC membranes, showing good adherence with the support (Figure 7). The Au-deposited thin-film follows the porous microstructure of the membrane, and shows good conductivity.

#### 3.4.2. Detection of *B. thuringiensis* with the IDE Setup

Interdigital electrodes, which are among the most commonly used periodic electrode structures for fringing field detection [32], were used for impedance measurements in order to substantiate the parallel-plate measurement results. An important advantage of this electrode design is that only a single-side access to the test material is required.

Figure 8A shows a lower impedance modulus due to bacterial presence, with a resistive angle. This expresses the fact that bacterial solutions are sensed through a decrease in the solution resistance resulting from an increase in conductivity. Given the single-side access to the material, the conductive phenomena of the nitrocellulose soaked with PBS:1000 are not shielded by the isolating backing layer, resulting in highly resistive impedance phase over the 1kHz–1 MHz range.

The sensitivity towards 10^8^ CFU mL^−1^
*B. thuringiensis* cells was evaluated over the whole spectrum, and presents a plateau of >18% at 10 kHz–0.3 MHz, resulting essentially from the almost stable difference of the impedance modulus in this frequency range. Due to the very resistive nature of the phase over this range, the modulus remains quasi-independent of the frequency.

#### 3.4.3. Sensitivities towards *B. thuringiensis* Cells

The sensitivities towards 10^8^ CFU mL^−1^
*B. thuringiensis* cells in low salinity buffers were evaluated over the whole spectrum for both parallel-plate and IDE setups. The spectral sensitivities of the systems were computed from the complex impedance of the blank PBS:1000 buffer and the bacterial suspension (Figure 9).

Regarding the parallel-plate (Figure 9A), a maximal sensitivity of about 21% is observed around 30 kHz for the parallel-plate setup. The sensitivity computed from the complex impedance is superior to the modulus sensitivity (around 17% at 40 kHz) as it includes both contribution from the impedance modulus change and phase shift around this frequency (depending on the interplay between *R_NC_* and *C_NC_*). In this system, the sensitivity does not remain constant over the spectrum as *C_NC_* has a non-negligible contribution to the detection.

Contrary to the parallel-plate, the sensitivity of the IDE setup presents a plateau of >18% at 10 kHz–0.3 MHz (Figure 9B), resulting essentially from the almost stable difference of the impedance modulus in this frequency range. Due to the very resistive nature of the phase over this range, the modulus remains quasi-independent of the frequency.

## 4. Discussion

The main objective of this work was to accommodate electrical sensing on paper substrates towards simple, rapid, quantitative bacteria detection in aqueous solutions.

To this end, we first developed a biointerface by functionalizing the NC membrane with endolysin cell-wall binding domain (CBD) aimed at capturing bacterial cells through the whole membrane pore surface. The efficiency of CBD specific binding to *B. thuringiensis* was demonstrated in [28]. In this paper, we validated their potential as immobilized bioreceptors on porous NC membranes, taking advantage of their highly selective binding capacity to capture *B. thuringiensis* bacterial cells present in low-salinity buffers. As the activity of the endolysins was shown to decrease at salt concentrations higher than 200 mM [22], it is necessary to assess the binding capacity of the CBD biointerface towards bacteria under different NaCl concentrations. This assessment is required to determine the range of aqueous solutions that can be considered for bacteria detection without loss of the sensor specificity. Also, the specificity of the detection systems towards *B. thuringiensis* is usually assessed with the absence of electrical signals in the absence of bioreceptors or in the presence of only non-target pathogens. Therefore, the detection specificity remains to be studied, and will be the topic of future works. When considering specificity of the detection, it will be of upmost importance to use a complete lateral flow assay with controlled flow as support to perform the electrical detection and a related robust detection protocol integrating the necessary washing steps.

Second, we took advantage of the NC support to develop an innovative volume-based electrical detection setup by applying the membrane between the electrodes of a parallel-plate prober for dielectric measurements. This setup forms a simple plug-and-play sensing device which is responsive to the electrical properties of the NC membrane. A nitrocellulose membrane functionalized with a specific biointerface is indeed an adequate substrate to support the volume-based electrical detection of bacterial suspensions. Such a detection scheme is attractive as the whole sample volume contributes to the sensing, increasing the contribution from targeted bacteria with respect to surface-based methods. The parallel-plate detection scheme based on impedance spectroscopy proposed in this work was much more straightforward both in fabrication and handling than most of the paper-based electrical biosensors encountered in the literature. They often rely on direct current monitoring requiring to apply more complex electrode systems to the paper (usually three electrodes, for reference, working and counter electrode) and involving electrochemical reactions which are relatively complex to interpret in ionic medium [40,41]. These paper-based sensors usually rely either on the application of conducting pastes showing high electrode and contact resistance (generally requiring custom, hand-made electrical connections) [41,42], or more complex and expansive fabrication processes from the microelectronics industry.

Third, we quantified the presence of 10^7^ to 10^8^ CFU mL^−1^
*B. thuringiensis* in diluted physiological buffer (PBS diluted 1000×). An important feature is the rapidity of the sensing mechanism: the whole detection protocol lasts less than 5 min. Using a simple electrical model including the electrical properties of the NC membrane and the capacitive contribution of the polyester backing, the interplay of both capacitive and resistive properties of the electrolyte are observed. The detection mechanism, based on ion concentration, was determined by the parallel-plate characterization of NC membrane filled with different saline solutions, and helped us understanding the sensor differential response for various electrolyte solutions. Similarities can be drawn in the complex impedance difference between the saline solutions and for biological buffer with and without bacterial cells. In particular, close correspondence is observed between electrical properties of PBS:1000 and ~2 × 10^−4^ M NaCl solution, while signals from the 10^8^ CFU mL^−1^ bacterial suspensions are representative of those of the ~10^−4^ M condition. These results, supported by dielectric measurements, suggest that the electrical model proposed for the sensing of NC membranes soaked with ionic solutions can be extended to bacterial solutions. Furthermore, it also endorses the concept of bacteria detection through its surroundings ions, which has already been discussed in [25]: the differences in sensor response are attributed to the slight difference in ionic content, i.e., electrical conductivity, between sterile PBS:1000 and bacterial resuspension in PBS:1000. Indeed, centrifugation steps lead bacterial cells to release ions due to osmotic pressure and damaged cell walls [43].

Both for saline solutions and bacterial suspensions, an interesting phase shift occurs in the phase peaks between 10 and 1000 kHz, driven by changes in *f_3_* due to both *C_NC_* and *R_NC_* variations. Although the resistive character of the sensor is predominant (changes in R_NC_ affect the total module more than *C_NC_*), the possibility to monitor NC membrane property changes through variation of *C_NC_* renders this setup versatile in use and potentially more robust against ionic noise, strongly affecting *R_NC_*. To assess the relative advantage of the parallel-plate setup over traditional paper-based sensors, we also applied gold microelectrodes (IDE) to the NC membrane using a standard microfabrication process. The IDE sensor is only reactive towards *R_NC_*, and shows a high sensitivity to bacteria over a larger frequency range than the parallel-plate. Indeed, the electrode disposition does not prevent a direct conduction path between the two electrodes.

Despite its dependence on bacterial concentration, two reasons make the detection of bacteria through higher ionic content unsuitable for bacterial sensing. First, the resulting signal is strongly affected by experimental procedures such as manipulation, contamination and experimental conditions, all affecting the baseline sample conductivity. Second, such a sensing principle is useless for real applications dealing with detection in highly saline solutions, whose high electrical conductivity is hardly impacted by bacterial ion release. Thus, the detection of bacteria through ionic contribution has the main disadvantage that it lacks robustness in a complex environment with various living organisms that are potential ion-sources, resulting in a low signal-to-noise ratio (SNR). The specificity and sensitivity of the electrical detection can be improved by nanoparticles (NP) to specifically label whole bacterial cells. In [44,45,46], *E. coli* O157:H7 were detected through amplified conductance and permittivity changes by means of the conjugation of specific graphene or gold NPs to the bacteria. In other works, the highly intensive response of Si-NPs and Au-NPs conjugated to bioanalytes at radio frequencies (RF) was used to amplify the dielectric contrast of bioanalytes in solutions [47,48] or to act as microantenna in NC membranes [49].

The ability of the parallel-plate setup to monitor changes of both conductive and dielectric properties in the NC membrane makes it possible to select the type of NP (conductive or dielectric) that increases the SNR the most. Conjugating diverse types of NP with bacterial cells offers promising perspectives for highly specific electrical bacterial detection on lateral flow assays, and will be the focus of upcoming works.

## 5. Conclusions

In this paper, an innovative method for the electrical characterization of cellulose-based membranes was proposed towards the development of low-cost, quantitative paper-based biosensors. It consists in the use of a simple parallel-plate setup as sensor to perform impedance measurements on the membrane inserted within the electrodes. The sensing principle was studied and validated by detecting saline solutions of different molar concentrations spread within nitrocellulose membranes. We then demonstrated the proof-of-concept of the detection of bacteria: impedance-based detection of 10^8^ CFU mL^−1^ of *B. thuringiensis* cells was presented without labeling nor signal enhancement strategies. The bacteria were detected through an overall increase in ions in the membrane caused by their presence. Impedance measurements were also performed with interdigital electrodes integrated on the membrane and confirmed the parallel-plates results. Finally, newly discovered endolysin CBD were introduced as specific bioreceptors and deposited inside the nitrocellulose membrane, enabling successful bacterial cell capture over the whole membrane volume.

In conclusion, by combining the benefits of a cellulose-based membrane, novel protein bioreceptors and precise impedance measurements with a reusable plug-and-play setup, we obtained promising results towards the development of an affordable and sensitive biosensor with a speed of response under 5 min. To overcome the limitations presented by our system and reach the very low limit of detection required for example for drinking water assessment, it is necessary to integrate signal enhancement strategies. Future research works will explore the use of nanoparticles as labels to increase the analytical performances of the device. The development of such a versatile tool creates novel opportunities in situations that require rapid and frequent pathogen detection, such as the detection of *E. coli* in drinking water. Sensing applications could be extended to the detection of various pathogens and viruses as well, through appropriate and direct modification of the biointerface. This may prove particularly useful in emergency situations in light of the recent coronavirus disease 2019 (COVID-19) pandemic.

## Figures and Tables

**Figure 1 biosensors-11-00057-f001:**
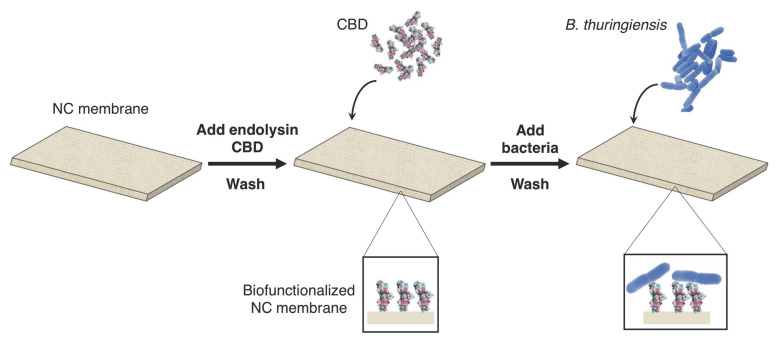
Protocol of the nitrocellulose (NC) membrane biofunctionalization with endolysin cell-wall binding domain (CBD) and validation of the biointerface through capture of *B. thuringiensis* cells. (1) The deposition of the phage endolysin CBD in the NC membrane is followed by drying, washing and desiccator steps. (2) The bacteria are then deposited on the membrane and specifically captured by the CBD.

**Figure 2 biosensors-11-00057-f002:**
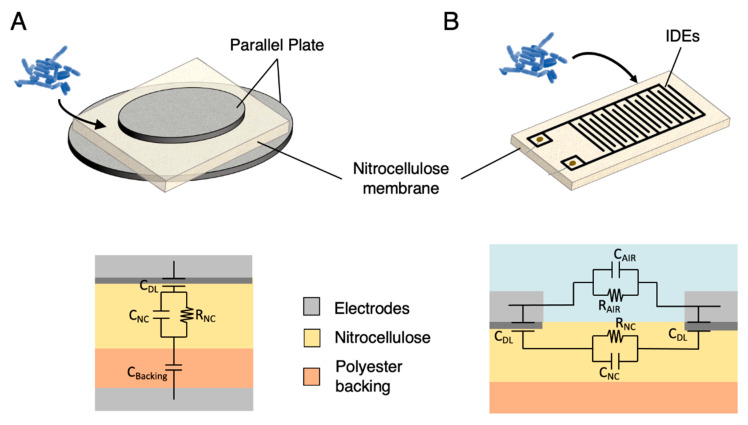
Experimental setups and corresponding electrical models investigated towards bacteria detection in this work. (**A**) Parallel-plate probes are a common material dielectric measurement system. The bacterial sample is deposited on the NC membrane and conducted to the test zone by capillarity. A simple electrical model is proposed to consider the dielectric effect of the polyester backing supporting the NC and the electrical double layer that arises from charge redistribution at the interface between the electrolyte and the probe. (**B**) Interdigital electrodes are generally used as sensors to monitor impedance changes at the proximity of the metallic fingers, here deposited on NC membrane. The bacterial samples are deposited on top of the interdigital electrode (IDE)-NC sensor. The model does not include the polyester backing as its impact on the impedance seen by the IDE is negligible due to its depth.

**Figure 3 biosensors-11-00057-f003:**
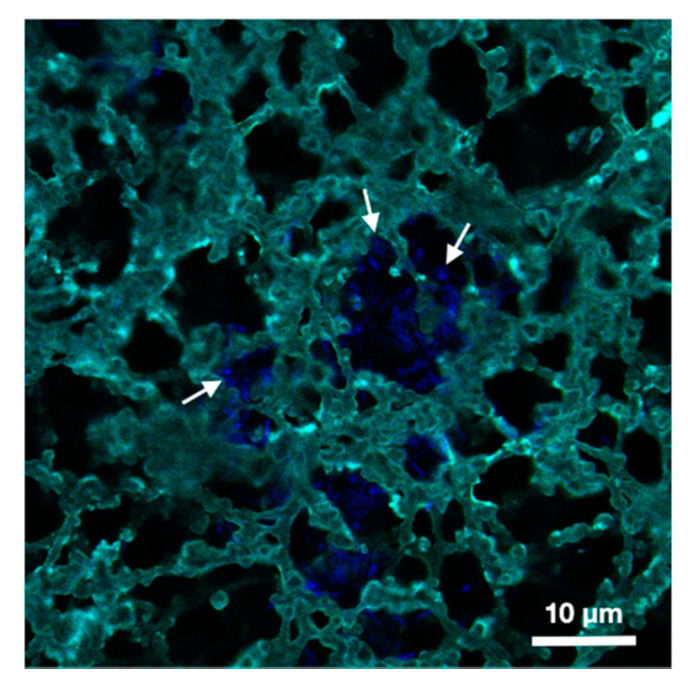
Confocal fluorescence microscopy image of immobilized *B. thuringiensis* cells (blue) in the pores of a CBD-biofunctionalized nitrocellulose membrane (turquoise). The white arrows indicate the presence of captured bacteria on the surface of the membrane pores.

**Figure 4 biosensors-11-00057-f004:**
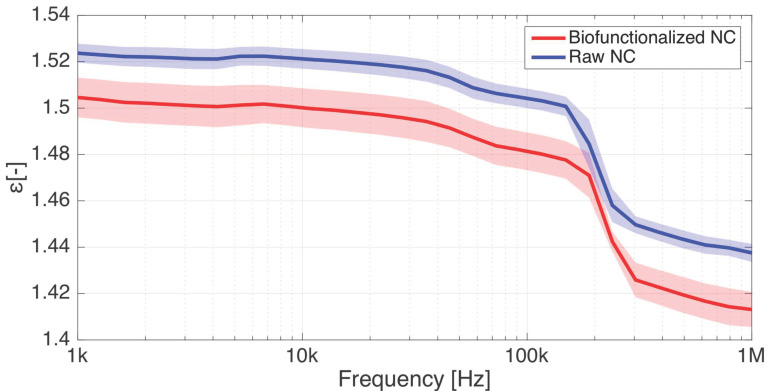
Relative permittivity of raw and CBD-biofunctionalized nitrocellulose membranes over 1 kHz–1 MHz. The biofunctionalization process causes a small decrease in the permittivity. Total number of samples: 7. Shaded area surrounding the measurement curves: standard error (σ).

**Figure 5 biosensors-11-00057-f005:**
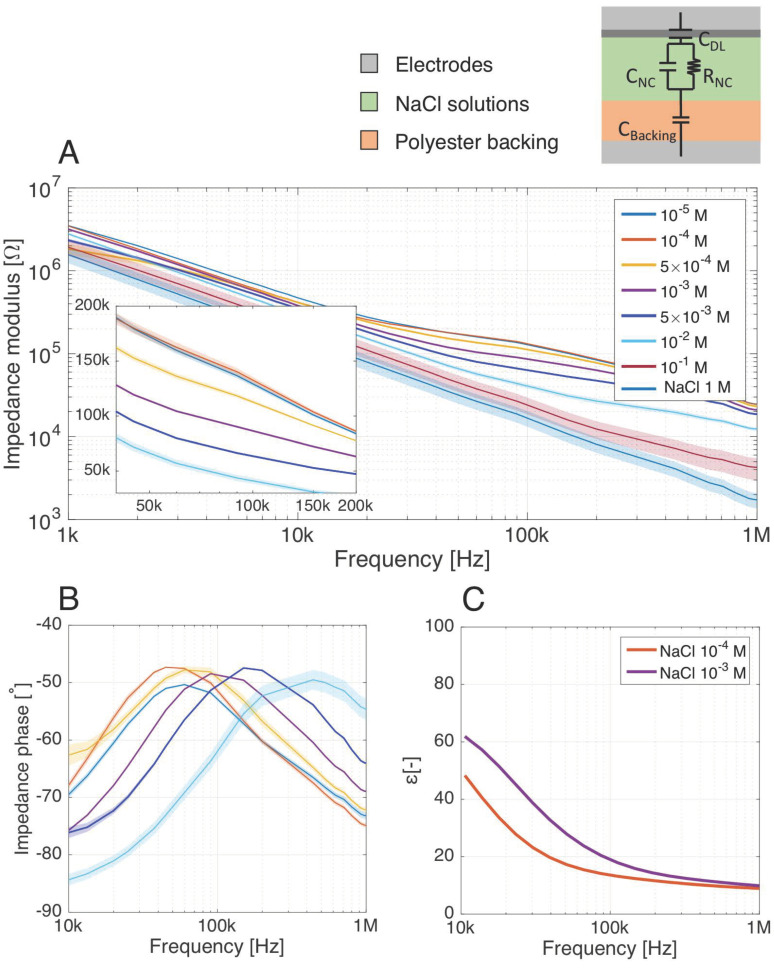
(**A**) Impedance measurements of the nitrocellulose (NC) membrane with isolating backing as seen by the parallel-plate setup. The NC membrane is saturated with saline solutions of different molar concentrations, modelling the dielectric properties of real water samples. A significant decrease in the impedance modulus results from an increase of the ionic strength, expressed as a drop in the membrane resistance *R_NC_*. (**B**) The impedance phase evolution over 10 kHz–1 MHz highlights the contribution of both *R_NC_* and *C_NC_* to the impedance of the system. As the resistive peak in the phase shifts with the ionic strength, the system is, therefore, sensitive to saline electrolyte through both *R_NC_* and *C_NC_* changes. (**C**) Dielectric relative permittivity of the system as seen by the parallel-plate setup, with different diluted salt solutions in the nitrocellulose membrane. Number of samples: 12 from two independent experiments. Shaded area surrounding the measurements curves: standard deviation (σ). σ smaller than the measurement line thickness if not visible on the graph.

**Figure 6 biosensors-11-00057-f006:**
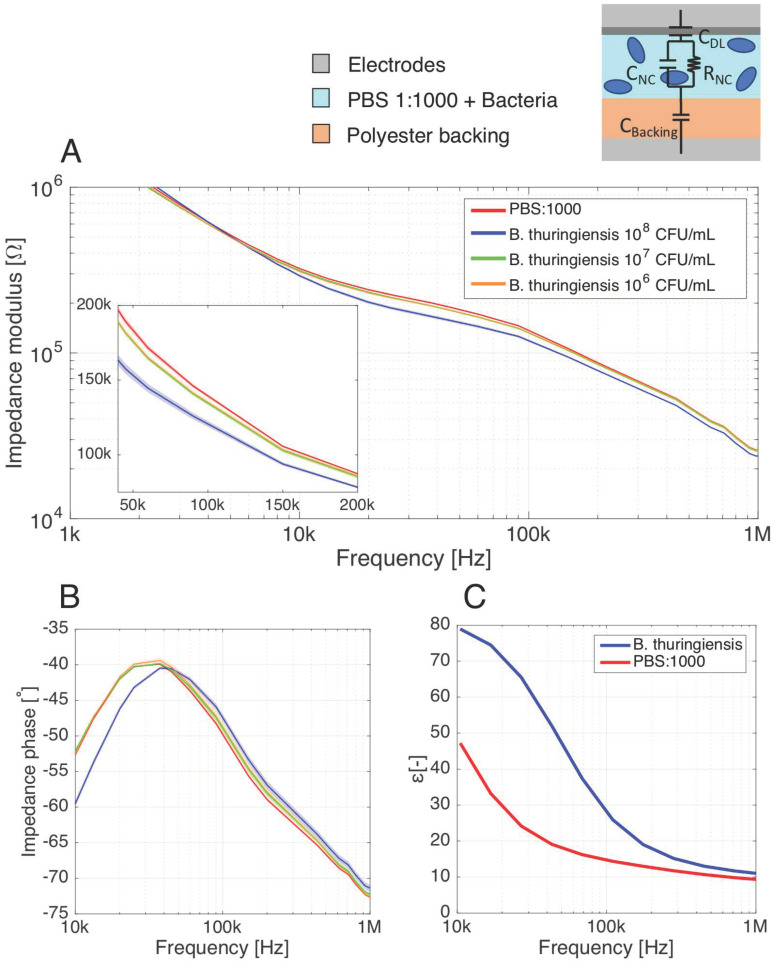
(**A**) Impedance measurements performed on a nitrocellulose (NC) membrane as seen by the parallel-plate setup. The NC membrane is saturated with phosphate-buffered saline (PBS):1000 (reference buffer) or with PBS:1000 containing 10^8^, 10^7^ and 10^6^ CFU mL^−1^
*B. thuringiensis* cells. The global impedance is experimentally shown to decrease, and the phase is subjected to a shift in presence of bacterial cells in the buffer, showing high similarities with the response of slightly saline solutions. (**B**) The impedance phase evolution over 10 kHz–1 MHz, presenting a phase peak, indicates an interplay of *R_NC_* and *C_NC_* in the impedance of the system when subjected to bacteria. (**C**) Dielectric permittivity of the system as seen by the parallel-plate setup, when subjected to low-salt buffer with and without bacterial cells. Number of samples: 12 from two independent experiments. Shaded area surrounding the measurements curves: standard deviation (σ).

**Figure 7 biosensors-11-00057-f007:**
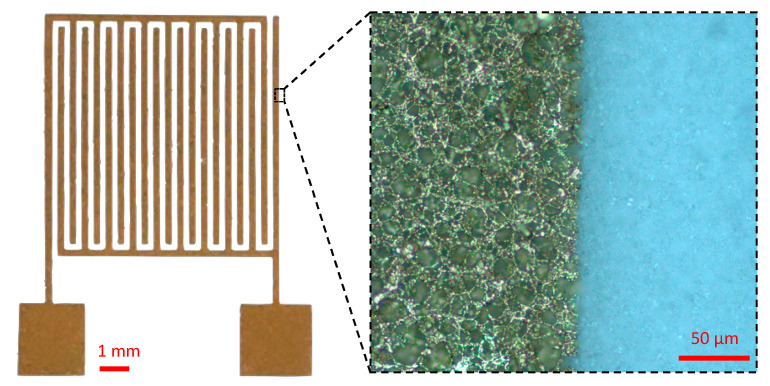
Optical microscopy images of Au-IDE (200 μm of interdigit gap) deposited on a nitrocellulose membrane. The inset is a zoom in image (magnification 20×) of the electrode showing the Au deposition on the membrane surface as well as in the first microns of the membrane thickness due to NC porosity.

**Figure 8 biosensors-11-00057-f008:**
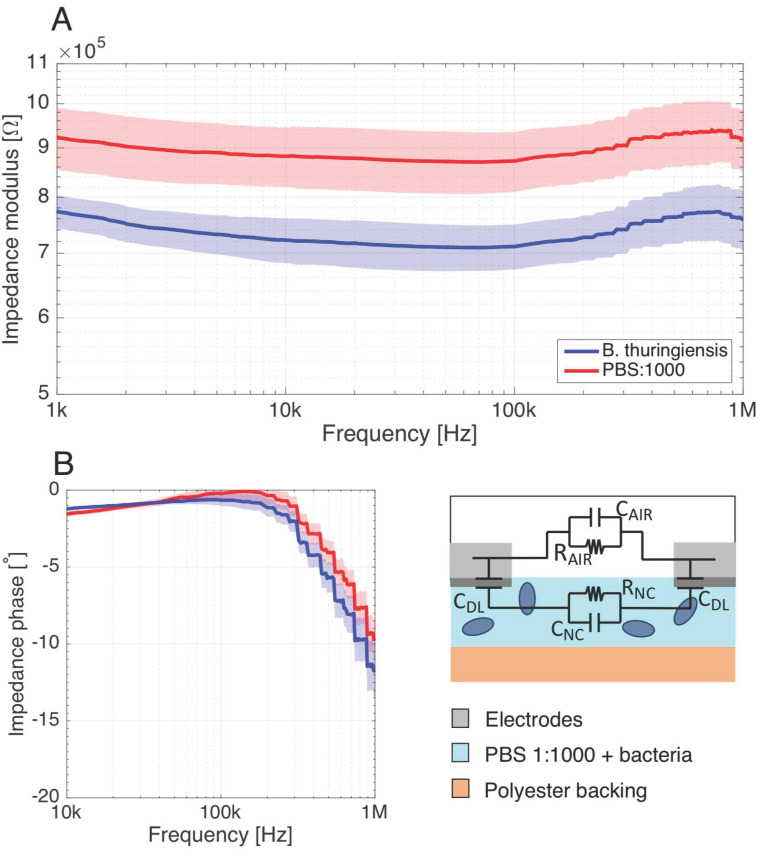
(**A**) Impedance measurements performed on nitrocellulose (NC) membrane as seen by the IDE, with and without bacterial cells in PBS:1000 buffers. The impedance modulus presents a constant impedance shift representing a decrease in the solution resistance. (**B**) Since a direct conduction path exists between their electrodes, the impedance seen by the IDE has a character that is rather resistive as expressed by the highly resistive phase over a large spectrum. Number of samples: 12 from two independent experiments. Shaded area surrounding the measurements curves: standard deviation (σ).

**Figure 9 biosensors-11-00057-f009:**
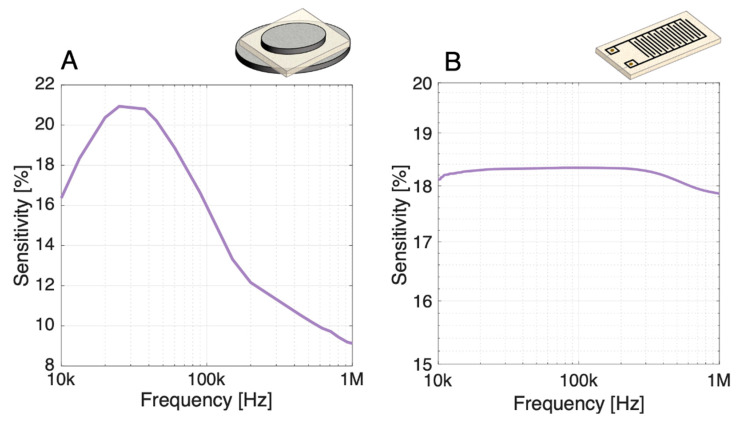
Spectral sensitivities of the (**A**) parallel-plate and (**B**) IDE setup towards 10^8^ CFU mL^−1^
*B. thuringiensis* cells in low salinity buffers. The sensitivity includes both contribution from the modulus and the phase shift. The IDE sensitivity expresses their high response towards resistance changes through a plateau between 10 kHz and 0.3 MHz, while the parallel-plate sensitivity presents a peak because of the interplay between *R_NC_* and *C_NC_* in the sensing mechanism.

**Table 1 biosensors-11-00057-t001:** Modelling of the electrical properties at 20 °C of real aqueous samples and biological buffers using saline solutions at different concentrations. Mean concentrations and, therefore, dielectric properties are considered as the physico-chemical content of the water samples can vary from place to place [34,35].

Modelled Solutions	*c_NaCl_* [M]	*ε_r,sol_* [/]	*σ_sol_* [S/m]
PBS:1000	1.6 × 10^−4^	~80	1.8 × 10^−3^
Drinking/surface water	~10^−3^	~80	10^−1^–10^−2^
PBS/highly saline water	1–5 × 10^−1^	~70	1–5

**Table 2 biosensors-11-00057-t002:** Monitoring of the nitrocellulose membrane resistance *R_NC_* and capacitance *C_NC_*, respectively showing consecutive relative decreases and increments with the concentration of the NaCl solutions used to model different types of water samples through their conductivities. The parallel-plate setup is more sensitive to *R_NC_*, but is still responsive to capacitance changes of the membrane.

	10^−4^ M	5 × 10^−4^ M	10^−3^ M	5 × 10^−3^ M	10^−2^ M	10^−1^ M
*ΔR_NC_*	1.9 × 10^4^ Ω	−15%	−35%	−47%	−57%	−90%
*ΔC_NC_*	8.34 pF	+5%	+17%	+15%	+35%	+22%

## Data Availability

The data presented in this study are available on request from the corresponding author.

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
