# Peer review of "Electrical Characterization of Cellulose-Based Membranes towards Pathogen Detection in Water†"

_biosensors, 2021, doi:10.3390/bios11020057_

Round 1

Reviewer 1 Report

The manuscript describes a nitrocellulose (NC) membrane based sensor for pathogen detection using EIS signal.

This is a beneficial attempt to develop a simple and rapid-response sensor for bacterial detection.

The study works and manuscript are both well organized. The results and related discussion sound to be reasonable.

Reviewer 2 Report

The manuscript by Le Brun et al. “Electrical Characterization of Cellulose-Based Membranes towards Pathogen Detection in Water” describes the development of sensing devices responsive to the electrical properties of the cellulose-based membrane. Precise analysis of impedance phase and modulus of the developed devices allows detecting B. thuringiensis cells. This proof-of-the- concept paper is timely and could be published after slight improvements of the introduction aimed to attract a more general reader.

P2 line 76. In this paragraph, authors describe pros and cons of electrical biosensors. I think, it might be slightly expanded as proposed below to attract the attention of a general reader. “Electrical biosensors, on the other hand, rely on the monitoring of changes in material electric properties when bacteria bind in close proximity from the surface of, e.g. interdigital electrodes (IDE) designed on a solid substrate [11-13]. The signal response, typically proportional to the number of bacteria, is used as an electrical fingerprint of the sample to provide fast, precise and quantified information about the bacteria presence in water.” I would suggest adding here: “However, particular mechanisms of electron transfer between electrodes and specific bacterial cells as well as within the cells is still under fundamental studies in bioelectrochemistry.” [I would suggest here referencing to recent review in Sensors 202020(12), 3517; https://doi.org/10.3390/s20123517] “However, in conventional surface-based electrical biosensors, grafting a biorecognition layer, e.g., bacteriophages or antibodies, on the sensor surface typically faces problems such as reproducibility, uniformity and stability over time [14], and the functionalization protocol needs to be adapted to every surface material and grafting molecule. In addition, the capture percentage of bacteria by the biorecognition layer is relatively low since only bacteria in close vicinity to the surface binds to the specific receptors. Many of the target pathogens thus are flowing over the electrode without binding, decreasing the sensor sensitivity.” I would add here the following: “Finally, conventional surface-based electrical biosensors utilize gold electrodes functionalized with bioreceptors/antibodies using classical thiol chemistry." [I would suggest citing here your Ref.11 M. Cimafonte, et. al., and a work by Kraatz group https://doi.org/10.1039/C6AY01978A  ]. And then add: "However, insufficient chemical stability of thiolates is one of the most serious problems for their applications in ambient and aqueous environments." [Please refer here to Vericat et al https://doi.org/10.1039/B907301A ] And  you come to paper-based electrodes and describe it advantages in detail.

Reviewer 3 Report

The manuscript describes paper-based electrical biosensing of 108 CFU mL-1 of B. thuringiensis.

The Authors validate the use of a simple parallel-plate setup, analyzed the sensor response and resolution to saline solutions at different salt concentrations. Bacterium detection results with the plug-and-play parallel-plate setup were compared with a planar fringing field electrode system, composed of interdigital electrodes directly applied on a single side of the nitrocellulose membrane.

The article is well structured and written concisely. I suggest the paper to be accepted with minor revision. I enlist my specific comments below.

  1. The Authors state, that the present manuscript is an extended version of the paper published in: Le Brun, G.; Hauwaert, M.; Leprince, A.; Glinel, K.; Mahillon, J.; Raskin, J.-P. Electrochemical Characterization of Nitrocellulose Membranes towards Bacterial Detection in Water. In Proceedings of the 1st International Electronic Conference on Biosensors, 2–17 November 2020. However, there are similar sentences and even paragraphs in the present and other, already published articles, which can be considered as self-plagiarism. The Authors should paraphrase the text in the current manuscript.
  2. The Authors state, that the main objective of this work is to accommodate electrical sensing on paper substrates towards simple, rapid, quantitative and specific bacterium detection in aqueous solutions. However, quantification and specificity cannot be considered to be achieved yet.

Lack of specificity: the endolysin cell-wall binding domain (CBD) has been used as specific bioreceptor. Does the endolysin CBD indeed specificly bind only to Bacillus thuringiensis? Have the Authors measured bacteria capturing without CBD? The level (or lack) of such non-specific binding has to be demonstrated in the manuscript.

Lack of quantification: how can a proof-of-concept of the detection of bacteria be presented only on the basis of a single concentration of B. thuringiensis injected (108 CFU mL-1)? To demonstrate concentration-dependent quantification features of the system, at least two (preferably more) B. thuringiensis concentrations should be applied.
